# Rapid assessment of hand reaching using virtual reality and application in cerebellar stroke

E. L. Isenstein[1,2,3]*, T. Waz[1], A. LoPrete[3,4,5], Y. Hernandez[2,6], E. J. Knight[2,7], A. Busza[2,8], D. Tadin[1,2,3]*

**1** Department of Brain and Cognitive Sciences, University of Rochester, Rochester, NY, United States of America, **2** Department of Neuroscience, University of Rochester School of Medicine and Dentistry, Rochester, New York, United States of America, **3** Center for Visual Science, University of Rochester, Rochester, NY, United States of America, **4** Center for Neuroscience and Behavior, American University, Washington, DC, United States of America, **5** Bioengineering Graduate Group, University of Pennsylvania, Philadelphia, PA, United States of America, **6** The City College of New York, CUNY, New York, NY, United States of America, **7** Division of Developmental and Behavioral Pediatrics, Department of Pediatrics, University of Rochester School of Medicine and Dentistry, Rochester, New York, United States of America, **8** Department of Neurology, University of Rochester Medical Center, Rochester, NY, United States of America, **9** Department of Ophthalmology, University of Rochester School of Medicine and Dentistry, Rochester, New York, United States of America

* Emily_isenstein@urmc.rochester.edu (IEL); dtadin@ur.rochester.edu (TD)

**Data Availability Statement:** All relevant data are within the paper and its Supporting Information files.

## Abstract

The acquisition of sensory information about the world is a dynamic and interactive experience, yet the majority of sensory research focuses on perception without action and is conducted with participants who are passive observers with very limited control over their environment. This approach allows for highly controlled, repeatable experiments and has led to major advances in our understanding of basic sensory processing. Typical human perceptual experiences, however, are far more complex than conventional action-perception experiments and often involve bi-directional interactions between perception and action. Innovations in virtual reality (VR) technology offer an approach to close this notable disconnect between perceptual experiences and experiments. VR experiments can be conducted with a high level of empirical control while also allowing for movement and agency as well as controlled naturalistic environments. New VR technology also permits tracking of fine hand movements, allowing for seamless empirical integration of perception and action. Here, we used VR to assess how multisensory information and cognitive demands affect hand movements while reaching for virtual targets. First, we manipulated the visibility of the reaching hand to uncouple vision and proprioception in a task measuring accuracy while reaching toward a virtual target (n = 20, healthy young adults). The results, which as expected revealed multisensory facilitation, provided a rapid and a highly sensitive measure of isolated proprioceptive accuracy. In the second experiment, we presented the virtual target only briefly and showed that VR can be used as an efficient and robust measurement of spatial memory (n = 18, healthy young adults). Finally, to assess the feasibility of using VR to study perception and action in populations with physical disabilities, we showed that the results from the visual-proprioceptive task generalize to two patients with recent cerebellar

**Funding:** EI received an award from the Child Neurology Foundation (https://www.childneurologyfoundation.org) to be used at EI's discretion for research. There was no grant number associated with this award. AB received funding from the National Institute of Child Health and Human Development (https://www.nichd.nih.gov) NTRAIN K12 program (NIH/NICHD 1K12 HD093427-01). The funders had no role in study design, data collection and analysis, decision to publish, or preparation of the manuscript.

**Competing interests:** The authors EI, AB, and DT would like to declare the following patent application associated with this research: "Rapid and Precise Assessment and Training of Motor Behavior in Stroke Patients," (Pending). This does not alter our adherence to PLOS ONE policies on sharing data and materials.

stroke. Overall, we show that VR coupled with hand-tracking offers an efficient and adaptable way to study human perception and action.

## Introduction

Head-mounted virtual reality (VR) provides a multisensory and engaging experience by immersing the user in a 360˚ computer-generated environment. This technology affords an opportunity to change the way that perception and action research is conducted, bringing the potential for tightly controlled yet naturalistic experiments that can be conducted while the participant is in motion. Historically, action-perception research has generally involved relatively rigid experimental setups where simple stimuli are presented, with participants indicating their perception with a button press. While this framework has led to major functional and mechanistic advances in our understanding of how the brain processes sensory stimuli, it often treats perception as a passive, unidirectional process and belies the complex reciprocity of the action-perception loop [1]. These experiments typically employ simple, two-dimensional stimuli and are conducted in quiet, confined spaces by stationary participants to achieve a high degree of experimental control [2]. Further, many studies involving movement tend to be restricted by a small number of reaching target locations [3–5] or the movement is limited to small actions such as pressing a button [6–8]. These limitations of typical perception and action experiments are motivating an effort to develop more active, naturalistic experiments [9–14]. The goal is to capture the dynamic, bidirectional richness and complexity of everyday experiences.

The promise of head-mounted VR displays is that they will allow us to conduct much needed naturalistic and interactive studies of human perception while giving up little, if any, of the experimental control that is the cornerstone of empirical perception research. With VR, we can undertake increasingly complex questions about perception while also applying the findings to more diverse populations in real-life contexts. Neuroimaging research has shown that human brains are more attuned to complex, naturalistic stimuli over those that are simple and artificial [15]. VR technology can be customized to present three-dimensional images [16–18], create the illusion of distant sounds [19,20], and provide haptic feedback to create engaging, multimodal stimuli that represent the lived experiences of research participants [21–23]. VR can also incorporate a high degree of control in a realistic and multisensory environment, ideal for high quality basic research. For example, a recent study used VR in conjunction with eye-tracking to progressively remove the color from peripheral vision during free-viewing of immersive 360˚ videos, dramatically revealing the limitations of human color perception in the visual periphery [24]. This technology has also been used to assess audiovisual speech perception in children [25] and verticality perception in patients with symptoms of dizziness [26].

VR environments can also be constructed to be responsive to user input, allowing participants to behave closer to how they would in a real-world situation [27–29]. This sense of 'presence', which captures the feeling that a user is truly *there* in virtual world, results from the immersion the user feels as a result of realistic multisensory illusions [30,31]. This feeling also provides a sense of agency over the environment, increases task engagement, and can affect cognition, social behavior, and memory [1,32,33]. Naturalistic stimuli also capture and maintain attention more authentically than simple two-dimensional stimuli because they tap into more sophisticated top-down attention pathways that incorporate context, prior knowledge, and goals rather than purely feature-based attention [34].

A recent benefit of head-mounted VR lies in its ability to easily capture data from a moving participant, allowing perception and action to be studied simultaneously during active, full-body tasks. As most research on perception is conducted with a stationary participant, this ability to concurrently examine how people physically interact with and respond to their environment provides new opportunities to study the action-perception loop. Further, some VR headsets are able to track the position of the hands in real time, including precise finger movements. One such device, the Oculus Quest (Meta, USA) has < 1 cm tracking accuracy in good environmental conditions [35]. The implications of simple and effortless body tracking technology are considerable; in particular, experiments studying human movement, posture, and proprioception in clinical populations stand to benefit from this technology. Crucially, the portability of VR headsets means that research can occur in places that cannot accommodate traditional lab equipment, such as a hospital room or out in the community. Larger groups of more diverse populations can be tested because conditions can be replicated with very high fidelity regardless of the participant's location or circumstances. Commercially available VR headsets are also impressively accessible in terms of cost, portability, and ease of use. As a portable "lab in the box," a headset has the potential to increase sample sizes, reach under-studied populations, and promote long-distance scientific collaborations.

One area of VR research that has received a great deal of attention is in stroke rehabilitation, with a specific focus on visual-motor coordination and perception. Over 100 randomized control trials have been conducted testing VR technology with people recovering from stroke, with the majority published in the past five years. There is substantial diversity in the attributes of the investigations: studies have been conducted in the home [36–38], in conjunction with telehealth resources [39–41], and in patients with both acute [42–44] and chronic [45–47] stroke. The majority of work on motor rehabilitation only assessed *gross* motor skills (e.g., reaching) by tracking the position of the handheld controller [44,48] or tracked finger motion by using supplemental specialty equipment [49,50]. However, persistent *fine* motor dysfunction is a common consequence of stroke and dramatically affects activities of daily living [51,52], requiring rehabilitative techniques that target fine motor skills. Hand-tracking technology built into VR offers a promising avenue to examine the speed, accuracy, and consistency of fine motor movements as baseline assessments and/or measures of rehabilitative progress.

To assess the feasibility of using VR technology to study fine motor skills in both healthy and clinical populations, the present study employed hand-tracking to measure accuracy in simple reaching tasks while varying multisensory and cognitive demands. This study was inspired by previous tasks that used mirrors [53] or tablets [54] to manipulate hand or target visibility during reaching. Two different experiments were conducted with healthy young adults: one assessed visual-proprioceptive integration versus isolated proprioception, and the other tested spatial memory. These two tasks were selected to examine the sensitivity of VR-based reaching assessment under different sensory and cognitive conditions. The visual-proprioceptive task was also completed by two individuals with recent cerebellar strokes to evaluate the practicality of successfully collecting this data with individuals with motor or vision difficulties. Overall, the goal of this study was to evaluate whether VR-based hand tracking can serve as a sensitive measure of differences in fine motor movements across various conditions in individuals with and without visuo-motor disabilities.

## Materials and methods

For Experiments 1 and 2, healthy young adult participants were recruited from the University of Rochester and the greater Rochester community. For experiment 3, two patients

rehabilitating from cerebellar strokes at Strong Memorial Hospital (Rochester, NY) were recruited. Each healthy participant completed the Edinburgh Handedness Inventory [55] and a demographic survey. All participants had normal or corrected to normal hearing and all healthy participants had normal or corrected to normal vision. Written informed consent was obtained from all participants as approved by University of Rochester Research Subjects Review Board.

The virtual reality experiments were conducted using a 1$^{st}$ generation head-mounted Oculus Quest running the latest OS/firmware at the time of testing. UNITY version 2019.4.2f was used to create the experiments. SideQuest, a free 3rd party software, was used with the scrcpy plugin (https://github.com/Genymobile/scrcpy) so experimenters could monitor what the participant saw on the headset during the experiment. Healthy participants were seated in the experiment room on a stationary chair whereas participants with recent stroke conducted the experiment in a stationary chair next to their hospital bed. All experiments were conducted with no objects in front of the participants in rooms with good lighting to optimize the environment for hand-tracking. All participants were given a brief introduction on how to navigate the virtual reality setup. Participants were instructed to keep their shoulders against the back of the chair during the entire experiment and were monitored continuously and given reminders as necessary. The Oculus Guardian system, intended to prevent actively moving users from exiting the designated 'safe' area by providing a visual warning when the user approaches the periphery of the Guardian area, was disabled to avoid disrupting the experiment. All participants were monitored continuously to maintain a safe experience. Participants were told to put the headset on and to adjust the straps so that it was comfortable. Those wearing corrective lenses were able to wear them under the headset. Help was offered if requested. Participants were also shown the inter-pupillary distance slider at the bottom of the headset, and told to move it around until they found their "sweet spot," where the images/text were clearest and most legible. The inter-pupillary distance on the Quest headset ranges from 58mm–72mm. This wide range allowed participants to adjust the lens spacing for a comfortable viewing experience in VR.

Once each experiment loaded, participants viewed a grey, featureless room. Instructions appeared directly in front of them, and rendered representations of each of their hands appeared. These hand renderings moved and articulated in real-time corresponding to the participant's real hand movements. Participants were asked to indicate which was their dominant hand; once a hand was selected, only that hand was visible and functional for the remainder of the experiment. To ensure the reaching distance was appropriate to the size and motor function of each individual, participants extended their dominant arm to calibrate the reaching distance before each experiment. The distance from the end of the extended arm to the headset was used as the distance of the radius on which target stimuli would appear.

Each healthy participant completed one practice session and two separate experiments, the Visible/Invisible Hand experiment and the Memory Delay experiment (see supporting information S1 and S2 Videos). Stroke patients completed one practice session and only the Visible/Invisible Hand experiment to reduce fatigue and avoid possible confounding cognitive factors in the Memory Delay experiment. In each trial of the practice session, a pink sphere (target) appeared along an invisible 60-degree arc at arm's length in front of the participant; the radius of this arc was set by the extended arm in the experiment's introduction and the arc extended indefinitely vertically. Using their dominant hand, participants were instructed to touch the target sphere with their index finger. Each trial ended when the fingertip passed through the arc; the target would then disappear and the next trial would begin regardless of the accuracy of the reach. They were then instructed to move their hand back to touch a cube that appeared just in front of their chest. The cube served as a reset point that appeared once

the target sphere disappeared. Once the index finger touched the cube, the cube would disappear and after 500 ms a new target sphere would appear randomly along the 60-degree arc. The program specifically recorded the difference in degrees between where the tip of the index finger passed through the arc and the center of the target, accounting for both horizontal and vertical error. Participants were encouraged to take breaks by resting their hands on their lap to avoid fatigue. Participants completed practice trials until they felt comfortable with the motions and the experimenter deemed them ready to begin the experiments.

The two experimental conditions retained the same basic structure as the practice session, but with two sets of key modifications.

### Experiment 1 –Visible/Invisible Hand

This experiment used the same introduction and structure as the practice session, but in 50% of the trials the rendering of the dominant hand became invisible during the reaching phase (Fig 1A and 1B). In these invisible hand trials, the participant had no visible feedback on where their hand was while they were reaching for the target, forcing high reliance on proprioception. The hand reappeared only after the reach movement was completed. Each participant completed 10 practice trials and 200 experimental trials. Experimental trials were split into 100 hand visible randomly interspersed with 100 hand invisible trials. For examples of both types of trials, see supporting information S1 Video. The experiment took between 5–6 minutes to complete in healthy adults.

### Experiment 2 –Memory Delay

This experiment used a similar introduction and structure as the practice session, but in 50% of trials we imposed a memory demand on the reaching task (Fig 2). 500 ms after the participant touched the reset cube, the target would appear and be followed by a tone 1200 ms later. The tone had a frequency of 440 Hz and a duration of 100 ms, and was set at a volume comfortably audible for each individual participant. The tone was presented bilaterally and acted as a cue for the participant to reach for the target location. In this experiment, the hand remained visible for the entire duration of the experiment. The critical manipulation was the visibility of the target before the reach. In 50% of the trials the target sphere would remain visible for the entire duration of the trial (Fig 2A). In the remaining 50% of the trials, the target sphere would only appear for 200 ms then disappear for the remaining 1000ms before the tone and remain invisible during the subsequent reach movement (Fig 2B), requiring the use of spatial memory to guide the reach. This approach mirrors established memory-guided reaching tasks by introducing a one second delay [56,57]. As in Experiment 1, participants completed 10 practice trials and 200 experimental trials. The program randomly interspersed the 100 trials in which the target sphere remained visible and the 100 trials in which the target sphere disappeared. For examples of both types of trials, see supporting information S2 Video. The experiment took 8–9 minutes to complete.

### Experiment 3 –Visible/Invisible Hand after cerebellar stroke

This experiment was identical to Experiment 1, except that the participants included two patients with recent cerebellar stroke. The only difference was that patients took between 15 and 20 minutes to complete the experiment.

### Statistical analysis

All experiments measured reaching accuracy of the dominant index finger by calculating the difference in degrees between the center of the target sphere and where the tip of the index

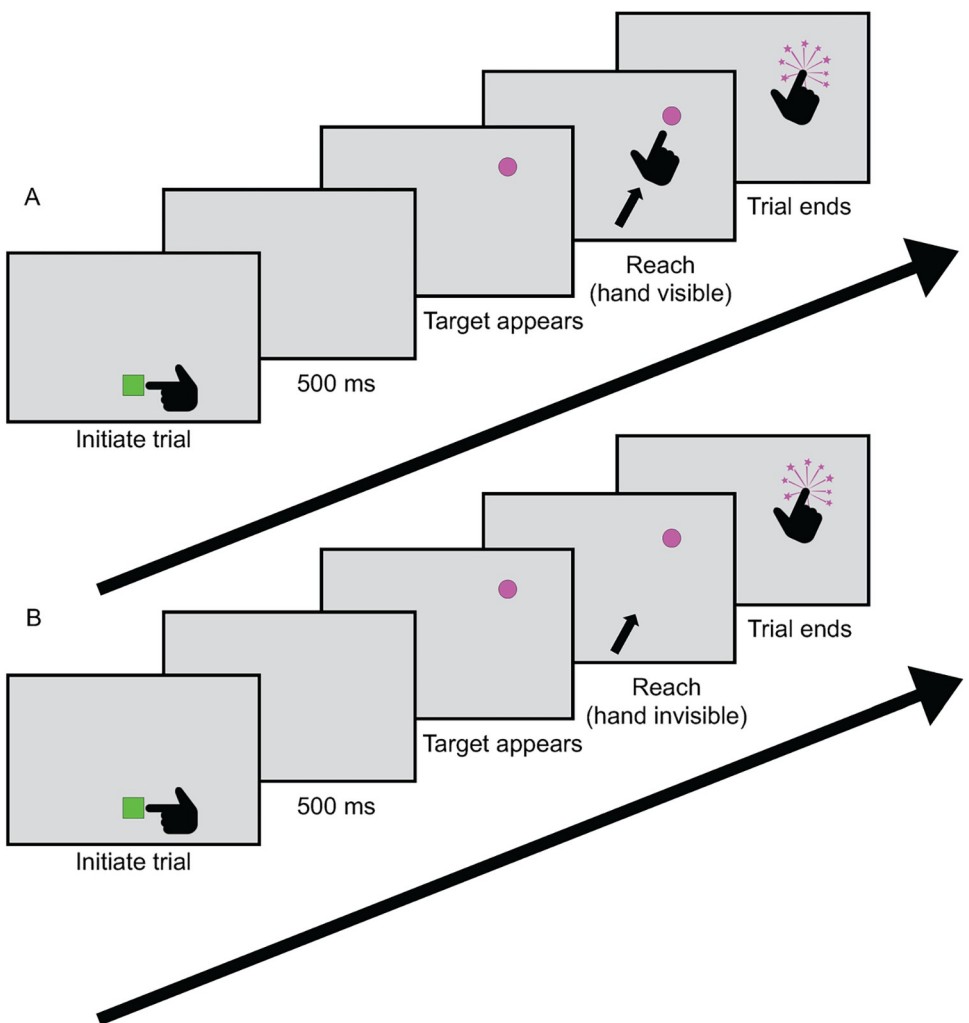

**Fig 1. Task and stimuli in the Visible/Invisible Hand experiment.** Each trial starts with a green cube appearing in front of the participant's chest. After the cube is touched, the cube disappears and a pink target sphere appears along a 60-degree arc in front of the participant at arm's length. When participant's index finger passes through the arc, it explodes and the trial ends. A new cube appears to begin the new trial. **A)** In the visible hand condition, the rendering of the hand is visible during the entire trial. **B)** In the invisible hand condition, the rendering of the hand is invisible during the reach phase. That is, the hand rendering disappeared when the cube was touched, reappearing only at the completion of the reach movement. For a video of this experiment, see supporting information S1 Video.

finger passed through any point along the 60-degree arc where the target could appear. This accuracy was compared between the two conditions of each experiment. In addition, each individual's precision was calculated as the standard deviation of their endpoint accuracy in Experiments 1 and 2. In Experiments 1 and 3, the reaching time–defined as the amount of time between when the target appeared and when the participant's index finger crossed the arc—was also recorded. This data is not available for Experiment 2. In all experiments, reaching accuracy was the main outcome measure as it has the greatest potential clinical significance and effect on quality of life and independence. Statistical testing was done with SPSS software version 28 (IBM Corp, Armonk, NY, USA) or MATLAB 2021a software (Mathworks, Natick, MA, USA). Shapiro-Wilk tests of normality were conducted on reaching time, accuracy, and precision in each condition in all experiments, with one or more conditions in each

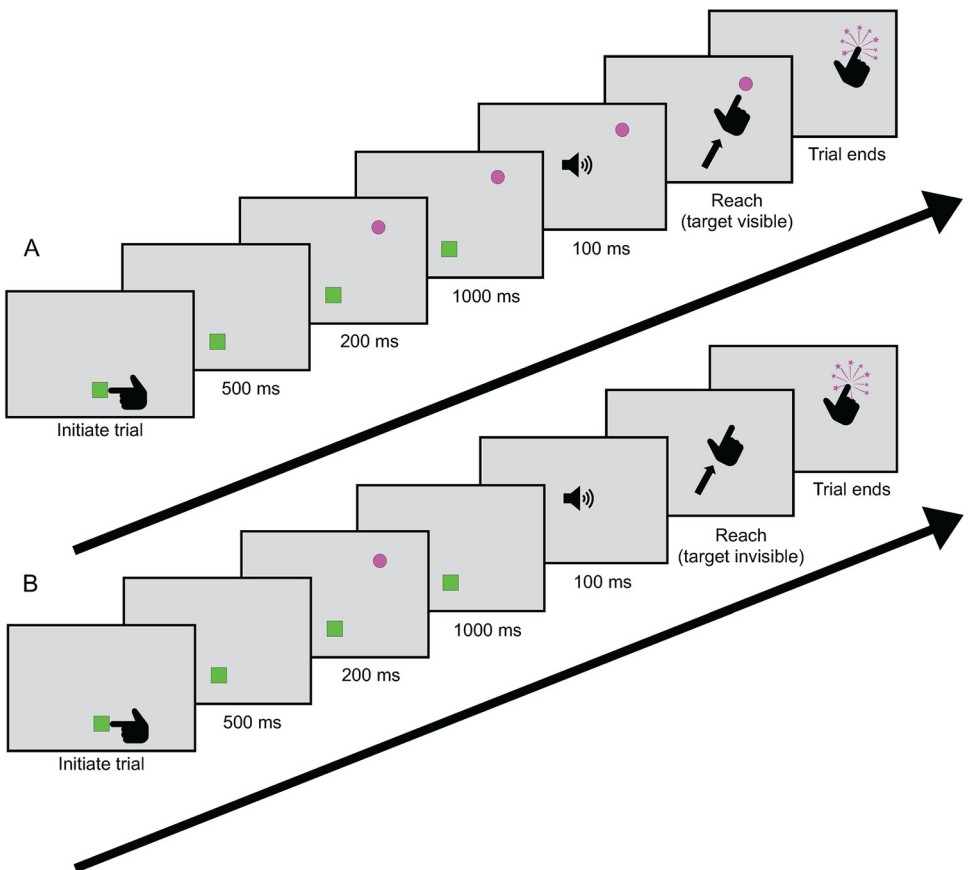

**Fig 2. Task and stimuli in the Memory Delay experiment.** Each trial starts with a green cube appearing in front of the participant's chest. 500 ms after the cube is touched, the pink target sphere appears along a 60-degree arc at arm's length. 1200 ms later, a tone indicates that a participant is free to reach out to the target. When participant's index finger passes through the arc, it explodes and the trial ends. A new cube appears to begin the new trial. **A)** In the standard condition, the target remained visible for the entire trial. **B)** In the memory delay condition, the target disappeared 200 ms after its appearance, remaining invisible for the 1000ms before the tone was played and during the subsequent reach movement. For a video of this experiment, see supporting information S2 Video.

experiment determined as non-normally distributed. Related-Samples Wilcoxon Signed Rank Tests were used in Experiments 1 and 2, as statistics were assessed on a group level. In Experiment 3, Independent Samples Mann-Whitney U Tests were conducted because statistics were assessed on an individual level. In Experiments 1 and 3, outliers of > 3 standard deviations away from each individual's mean were removed from the reaching time data. In Experiment 1, an average of 2.05 ± 1.00 outlier trials in the visible condition and 2.30 ± 1.26 trials in the invisible condition were removed per participant. In Experiment 3, 7 outlier trials in the visible condition and 2 in the invisible condition were removed for patient 1, and 6 outlier trials in the visible condition and 8 in the invisible condition were removed for patient 2. In all three experiments, reaching accuracy was also assessed including data from only the first 25 trials to test whether our approach is sensitive enough to detect the main results in substantially abbreviated versions of our experiments. Slopes of the change in reaching accuracy over time across conditions were normal across experiments; one sample t tests were conducted to assess whether the slope of the average error differed from zero. No power analyses were conducted prior to data collection because no suitable previous work was available to estimate the sample size needed.

**Table 1. Descriptions of patients included in recent stroke cohort.**

| | Age (years) | Time since stroke at time of participation (days) | Type of stroke | Level of motor/visual disability at time of participation |
|---|---|---|---|---|
| Patient 1 | 72 | 10 | • Large ischemic infarct in right cerebellum | • No muscular weakness, severe ataxia in right arm/leg<br>• Reported horizontal diplopia |
| Patient 2 | 75 | 1 | • Multifocal ischemic strokes, including large infarct in right cerebellum and right occipital lobe | • No muscular weakness, mild ataxia in right arm/leg<br>• Right homonymous hemianopia |

## Results

Twenty participants, 8 male and 12 female, participated in Experiment 1, with a mean age of 23.4 (st. dev. = 2.6). Eighteen of these participants, 8 male and 10 female, also participated in Experiment 2, with a mean age of 23.6 (st. dev. = 2.7). Information on the two patients rehabilitating from recent cerebellar stroke is found in Table 1. All participants, including patients, were right-handed, and reported no developmental or psychiatric disorders.

### Experiment 1 –Visible/Invisible Hand

The virtual hand experiment elucidated a clear, robust difference in the reaching accuracy when the virtual rendering of the hand was visible compared to when it was invisible (Fig 3A and 3B). We found a significant difference between the average reaching error in visible (2.24˚ ± .25˚) and invisible (3.80˚ ± .19˚) hand conditions (T = 204.00, z = 3.70, p < .001; Fig 3A). This difference was observed in a large majority of individual participants (Fig 3B). There was also a significant difference between the average reaching precision in visible (1.58˚ ± .76˚) and invisible (1.93˚ ± .69˚) hand conditions (T = 160.00, z = 2.05, p = .04). Precision and accuracy were shown to be positively correlated for both the visible (r(18) = .708, p < .01) and invisible (r(18) = .49, p = .02) hand conditions. There was no significant difference between the average reaching times in visible (625 ms ± 105 ms) and invisible (617 ms ± 160 ms) hand conditions (T = 87.00, z = -.67, p = .50).

To determine the sensitivity of this experiment at capturing differences in reaching accuracy, we repeated these statistical tests with only the first 25 trials of each condition. The difference between the visible (2.44˚ ± .37˚) and invisible (3.39˚ ± .52˚) hand reaching accuracy remained significant (T = 199.00, z = 3.51, p < .001). This finding, displayed in Fig 3C and 3D, confirms that the length of this experiment could be reduced to a fraction of the original length and still provide the same reliable, highly significant result in healthy adults. Participant level data is shown in Fig 4 to demonstrate the robust consistency of this data across participants and across the duration of the experiment.

To measure the stability of task performance over time and detect possible learning or fatigue effects, we assessed whether reaching accuracy results in either condition changed throughout the course of the experiment. On a group level, the slope of the average error was not significantly different from zero in both the visible hand (m = .002, std dev = .01, $t_{19}$ = .93, p = .36) and the invisible hand condition (m = -.0005, std dev = .01, $t_{19}$ = -.27, p = .79). Evidently, performance remained steady over the course of the full experiment, implying that there was no measurable learning or fatigue effects.

### Experiment 2—Memory Delay

The results of the Memory Delay experiment followed the same pattern as the Visible/Invisible Hand experiment, though results were slightly less robust. We found a significant difference

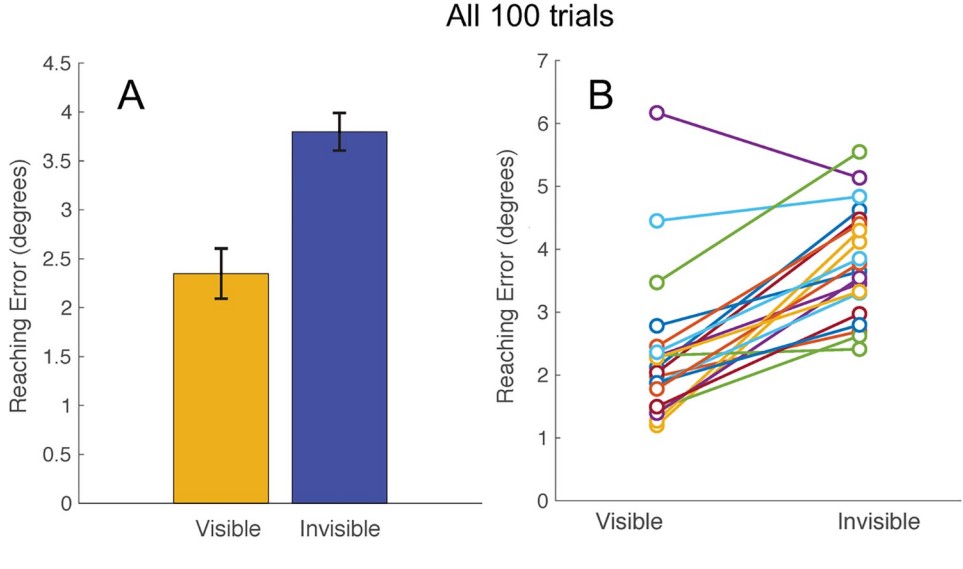

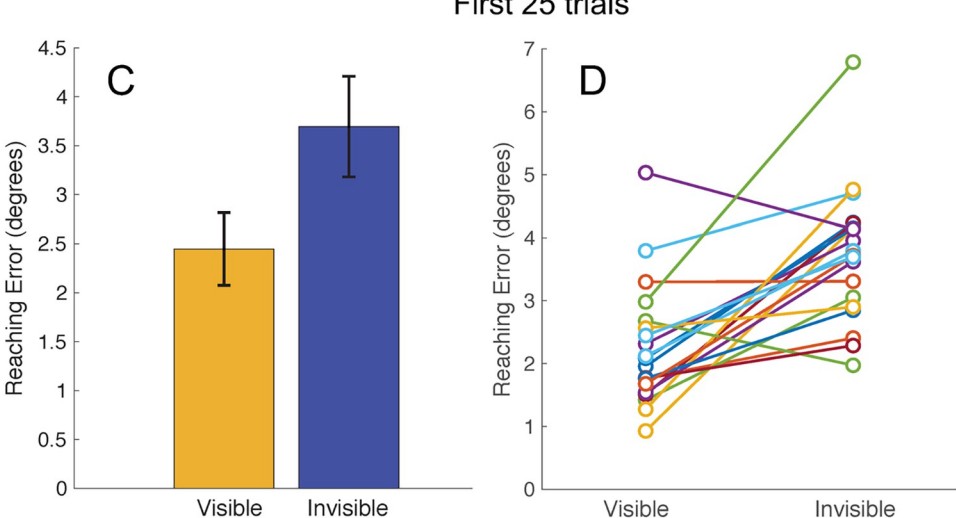

**Fig 3. Results of the Visible/Invisible Hand experiment in healthy adults. (A)** Group-level average reaching error as a function of hand visibility in all 100 trials. Yellow: Visible-hand condition. Blue: Invisible-hand condition. Error bars denote the standard error of the mean. **(B)** Results for 20 individual participants as a function of hand visibility in all 100 trials. **(C)** Group-level average reaching error as a function of hand visibility in the first 25 trials. **(D)** Results for 20 individual participants as a function of hand visibility in the first 25 trials.

between the average reaching accuracy error in the non-delayed standard condition (2.28˚ ± .27˚) and delayed target condition (3.45˚ ± .32˚; (T = 170.00, z = 3.68, p < .001; Fig 5A). Individual participant data is shown both as averages (Fig 5B) and with all trials shown (Fig 6). There was a significant difference between the average reaching precision in standard (1.47˚ ± .70˚) and delayed target (3.36˚ ± 3.54˚) conditions (T = 155.00, z = 3.03, p < .01). Precision and accuracy were shown to be positively correlated for both the standard (r(16) = .48, p = .04) and the delayed (r(16) = .69, p < .01) conditions.

Additional testing including only the first 25 trials continued to yield significant differences between the standard (2.00˚ ± .21˚) and delayed (3.37˚ ± .92˚) target conditions with respect to reaching accuracy (T = 158.00, z = 3.16, p < .01). Fig 5C and 5D demonstrate this robust

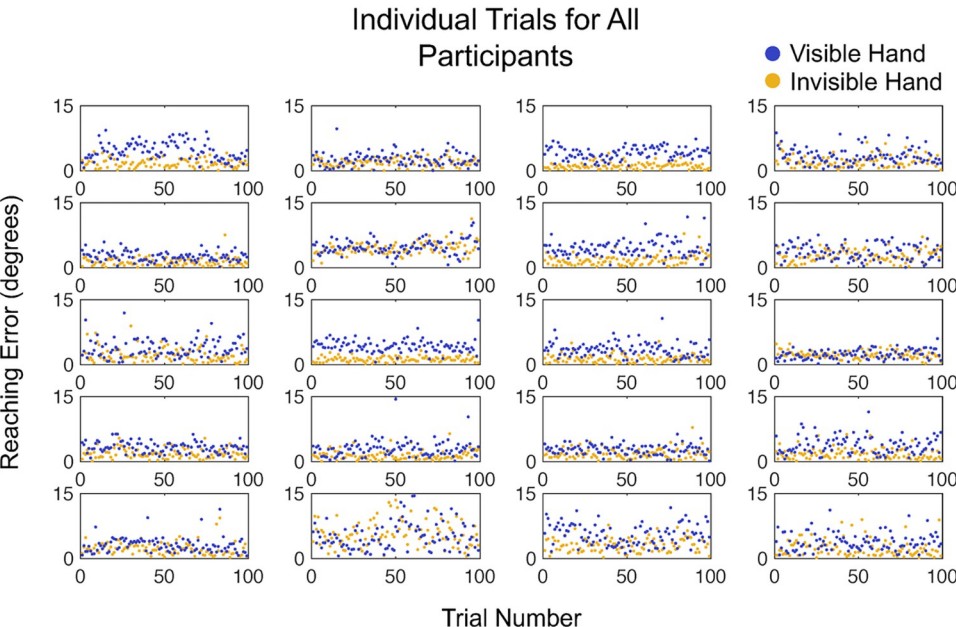

**Fig 4. Reaching errors for each individual trial in 20 healthy adult participants in the Visible/Invisible Hand experiment.** This depiction of the data allows for visualization of data stability over the course of the experiment. Yellow: Visible-hand condition. Blue: Invisible-hand condition.

finding after only a quarter of the total trials, affirming that the length of the total experiment could be substantially shorter the original and still reliably distinguish between trial conditions.

We again tested whether reaching accuracy in the two conditions changed over the course of the experiment to evaluate whether there were any learning or fatigue effects. On a group level, the slope of the average error was not significantly different from zero in both the standard (m = .0019, std dev = .01, $t_{17}$ = -.006, p = .464) and delayed condition (m = .000674, std dev = .02, $t_{17}$ = .45, p = .773). Thus, as with the first experiment, there were no significant changes in accuracy over time.

## Experiment 3—Visible/Invisible Hand after cerebellar stroke

We focused on the Visible/Invisible Hand experiment in patients with recent cerebellar strokes because the multisensory visual-proprioceptive interaction emphasizes body coordination, which is often affected by stroke [52]. This also minimized testing burden for the patients, who completed the experiment with their affected hands. In both patients, we found clear differentiation of reaching accuracy with and without assistance of vision (Fig 7A and 7C). Significant differences between the average reaching error in the visible (patient 1: 5.23˚ ± 2.17; patient 2: 3.49˚ ± 2.41˚) and invisible (patient 1: 8.94˚ ± 3.47; patient 2: 7.56˚ ± 2.60˚) hand conditions were found on an individual level: patient 1 U($N_{visible}$ = 99, $N_{invisible}$ = 99) = 3872.00, z = -2.55, p = .01; patient 2 U($N_{visible}$ = 99, $N_{invisible}$ = 99) = 8053.00, z = 7.82, p < .001. There were significant differences between the average reaching times in visible (patient 1: 1781 ± 1270 ms; patient 2: 4339 ± 6066) and invisible (patient 1: 1475 ± 916 ms; patient 2: 2922 ± 2145 ms) hand conditions (patient 1: U($N_{visible}$ = 94, $N_{invisible}$ = 98) = 3724.50, z = -2.29, p = .02); patient 2: U($N_{visible}$ = 95, $N_{invisible}$ = 92) = 3615.000, z = -2.04, p = .04).

We again assessed reaching accuracy after only 25 trials for each individual patient. The difference between the visible (patient 1: 5.63˚ ± 1.21; patient 2: 4.98˚ ± 3.23˚) and invisible

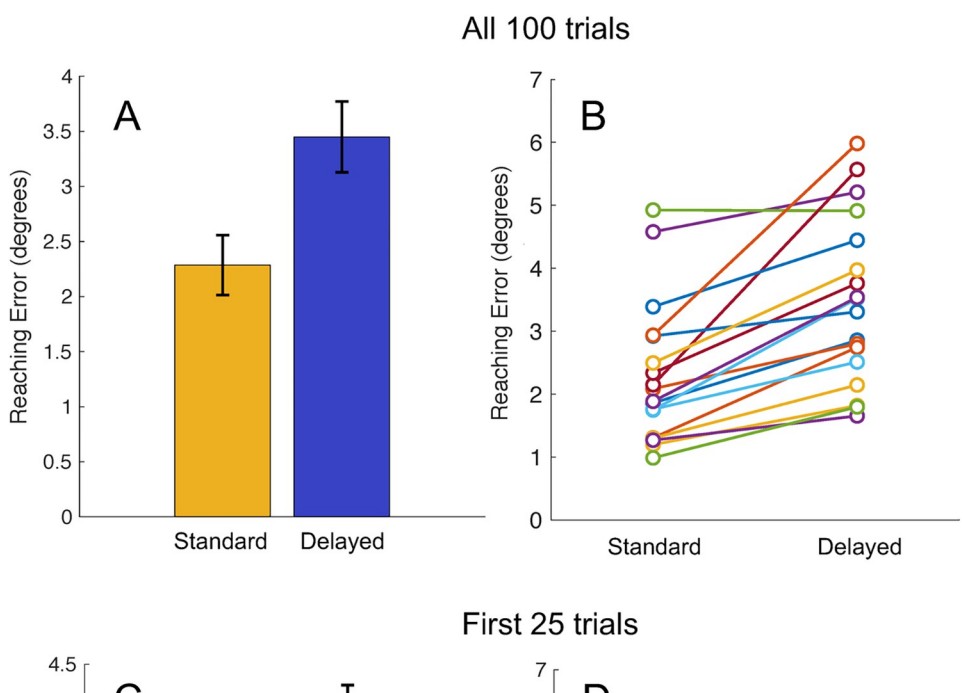

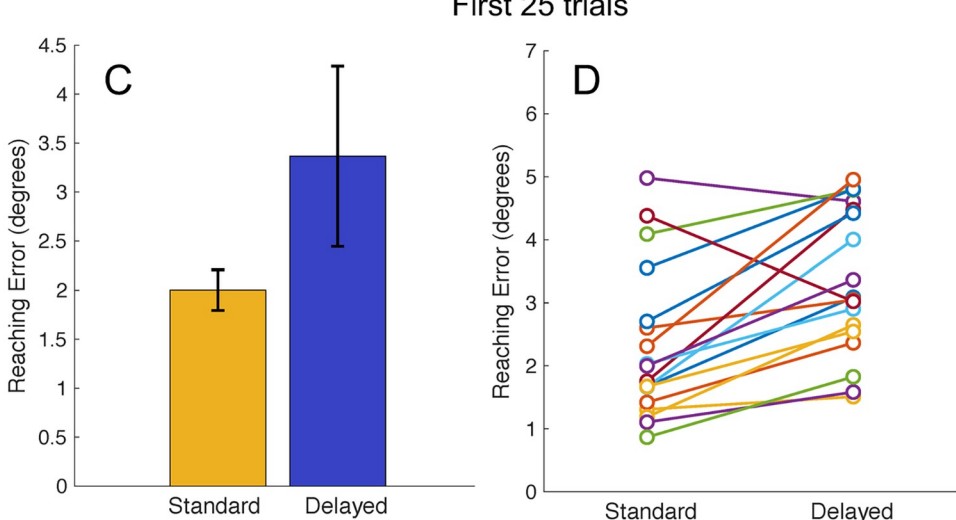

**Fig 5. Results of the Memory Delay experiment in healthy adults. (A)** Group-level average reaching error as a function of memory demand in all 100 trials. Yellow: Non-delayed standard condition. Blue: Delayed condition. Error bars denote the standard error of the mean. **(B)** Results for 18 individual participants as a function of memory demand in all 100 trials. **(C)** Group-level average reaching error as a function of memory demand in the first 25 trials. **(B)** Results for 18 individual participants as a function of memory demand the first 25 trials.

(patient 1: 11.66˚ ± 3.39˚ patient 2: 6.89˚ ± 3.18˚) hand reaching accuracy was significant: patient 1 U($N_{visible}$ = 25, $N_{invisible}$ = 25) = 605.00, z = 5.68, p < .001; patient 2 U($N_{visible}$ = 25, $N_{invisible}$ = 25) = 418.00, z = 2.05, p = .04 (Fig 7B and 7D). Participant level data is shown in Fig 8.

   Given the weakness and fatigue associated with cerebellar stroke, we evaluated the slope of the reaching error over time in each individual participant to assess for changes in accuracy over the course of the experiment. To determine statistical significance, we performed a bootstrap analysis in which we generated 10,000 bootstrap data sets. In each data set, trials were randomly resampled without replacement, thus retaining the overall distribution of the results but eliminating any temporal patterns of performance. This allowed us to assess the probability

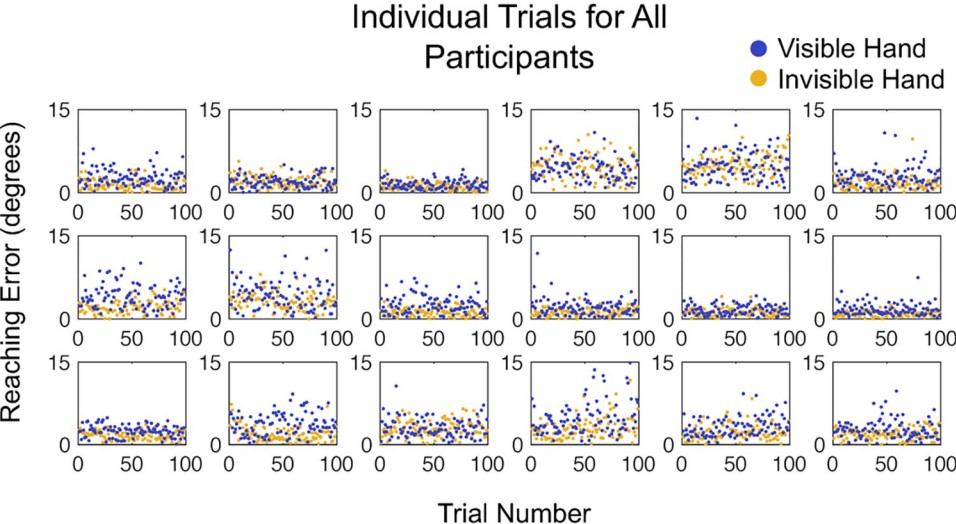

**Fig 6. Reaching errors for each individual trial in 18 healthy adult participants in the Memory Delay experiment.** This depiction of the data allows for visualization of data stability over the course of the experiment. Yellow: Non-delayed standard condition. Blue: Delayed condition.

that the observed slopes (Fig 8) differed from zero. In the visible hand condition, patient 1 had a slope of -.024 (p = .002) and patient 2 had a slope of -.025 (p = .001)—both showing significant improvement in performance over time. In the invisible hand condition, patient 1 had a negative slope of -.055 (p = < .0001) and patient 2 had a positive slope of .019 (p = .042). These findings show a mix of improvement and worsening that may reflect a learning effect or fatigue throughout the experiment.

## Discussion

Our results provide early evidence for the utility of built-in hand tracking in head-mounted VR equipment to quickly capture precise information about reaching accuracy. We were able to establish a significant faciliatory effect of vision on reaching accuracy (Fig 3) and demonstrate that adding memory demands impairs reaching accuracy (Fig 5). Our findings that people reach more accurately and precisely, though not more quickly, toward a point when they can see their hand and when the target is visible are not surprising. They confirm earlier data that vision improves accuracy and precision during reaching [58,59] and that reaching accuracy and precision deteriorate when memory is required to locate the target [60,61]. Rather, the novelty of the methods outlined in this paper lies in the manipulation of the sensory experience beyond what is possible in physical reality while collecting robust, consistent data anywhere in a matter of minutes.

By controlling the visual feedback provided by the hand rendering, we are able to uncouple vision and proprioception in the Visible/Invisible Hand experiment, offering a window into how these sensory modalities interact. Typically vision and proprioception are difficult to tease apart without the use of complex equipment such as mirrors [62] and robotics [63], but the use of this new VR technology allows for easy and modifiable adaptations. For example, instead of removing the visual representation of the hand, the rendering of the hand could instead be delayed or shifted to a different location to measure how these changes influence the weighting of visual and proprioceptive information. This weighting remains poorly understood in various clinical populations–such as cerebral palsy [64,65], Parkinson's disease

## Patient 1

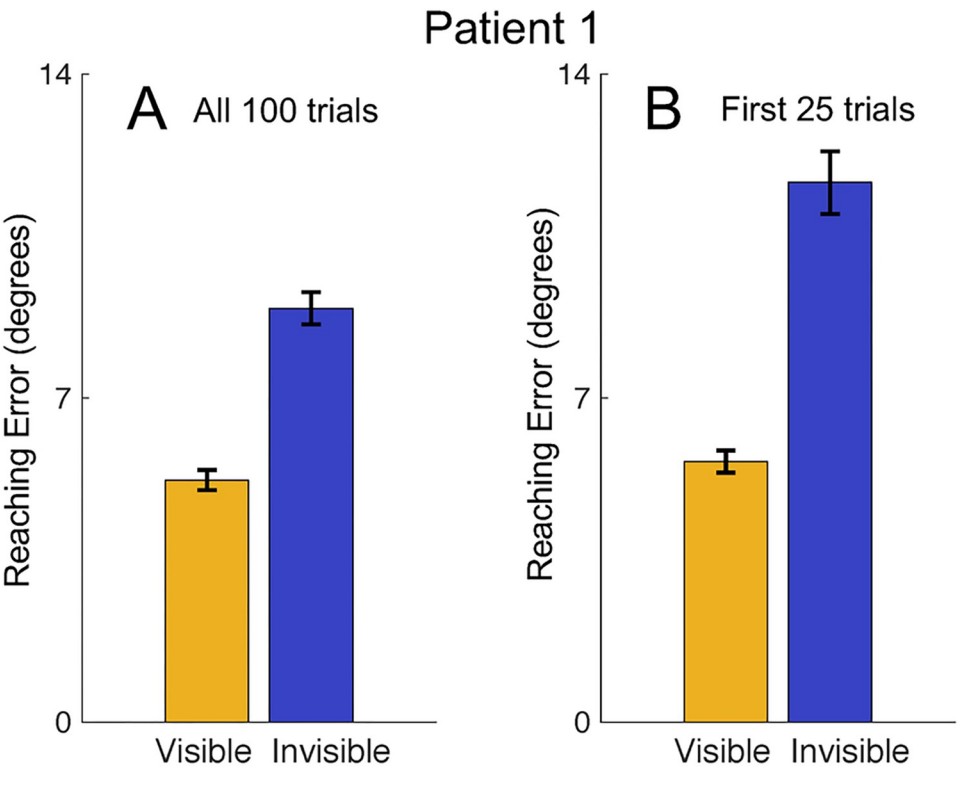

## Patient 2

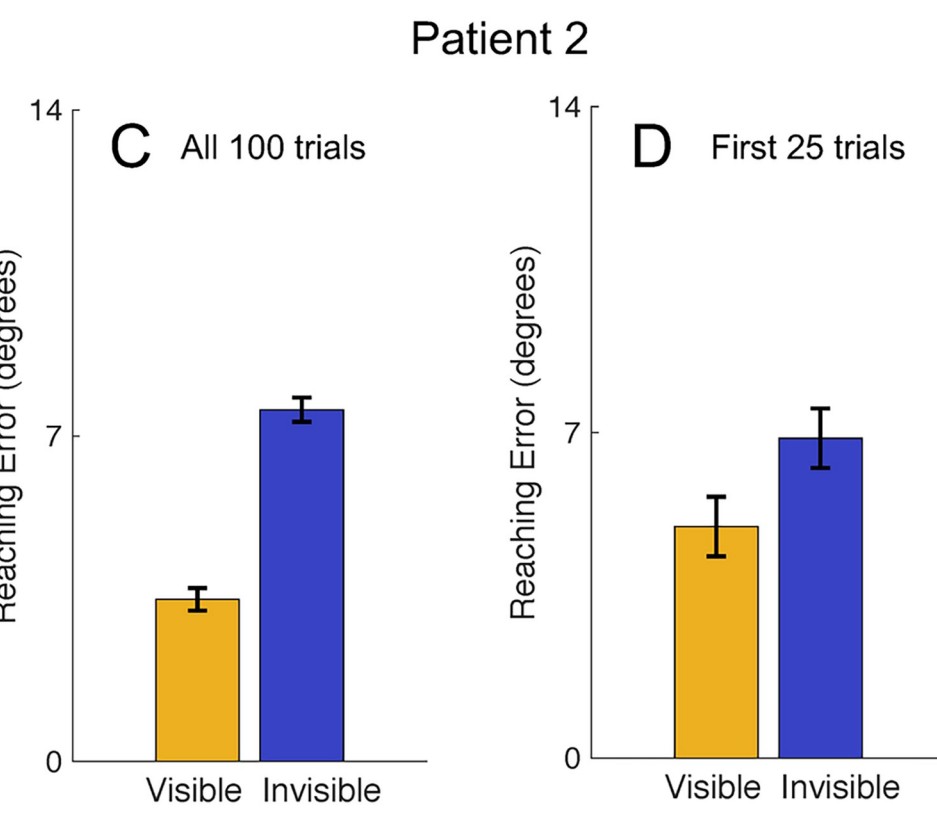

**Fig 7. Results of the Visible/Invisible Hand experiment in patients with recent cerebellar strokes.** (**A**) Reaching error as a function of hand visibility in all 100 trials in patient 1. Yellow: Non-delayed standard condition. Blue: Delayed condition. Error bars denote the standard error of the mean. (**B**) Reaching error as a function of hand visibility in the first 25 trials in patient 1. (**C**) Reaching error as a function of hand visibility in all 100 trials in patient 2. (**D**) Reaching error as a function of hand visibility in the first 25 trials in patient 2.

[66,67], and autism spectrum disorder [68,69]–that will benefit from research that can isolate and analyze the contributions of each sense and how they change over time.

By introducing a delay and requiring the participants to conduct their reaching movements based on recall, the Memory Delay experiment further assesses reaching in circumstances that require greater cognitive resources. While the delay in this paradigm was relatively short at 1 second, it still has a clear effect on the reaching accuracy. While this effect of memory is expected, our approach offers a way to investigate the spatial representation of memory in a three-dimensional setting. The environment can remain tightly controlled while objects are manipulated, allowing for structured and replicable assessments of spatial memory and navigation. Populations such as older adults and people with recent traumatic brain injury will benefit from further research on the interaction between memory and the ability to navigate a three-dimensional space [70,71].

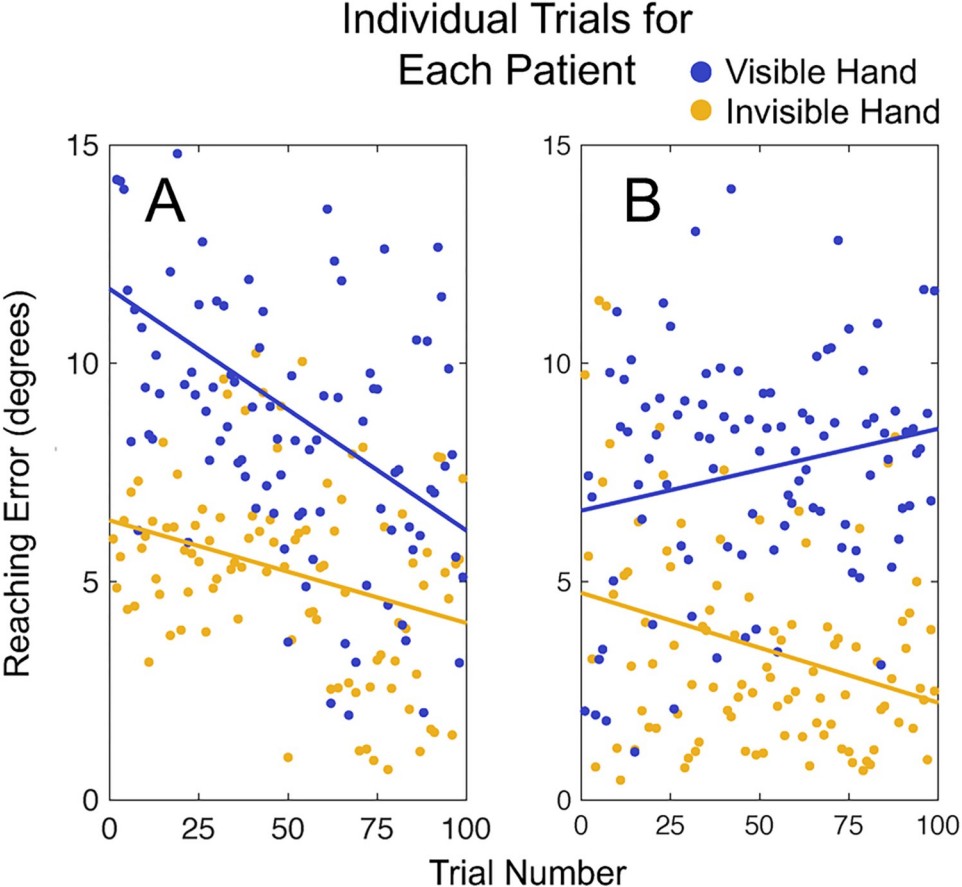

**Fig 8. Reaching errors for each individual trial in two patients with recent cerebellar stroke.** This depiction of the data allows for visualization of data stability over the course of the experiment. Yellow: Visible-hand condition. Blue: Invisible-hand condition.

Our study also contributes to decades of research confirming benefits when multisensory information is available in domains as varied as memory [72], learning [73], and reaction time [74]. In validating the use of VR to study multisensory processes, this new technique provides the capacity to expand on these traditional paradigms to evaluate participants as they move interactively with their environment. Overall, this approach allows for the measurement of action-perception data in a multisensory, naturalistic setting that can be adapted to mimic a variety of real-life scenarios better than the simple and predictable conditions typically found in the lab.

Critically, these experiments also show that VR can be used to efficiently and effectively measure reaching accuracy not only in healthy individuals, but also in those with vision or motor disabilities caused by cerebellar infarct. The self-paced nature of these experiments means that they can be adapted to suit individuals with limited mobility, and the ability to adjust the inter-pupillary distance and head position allows for reasonable correction of minor visual issues, as done with the first patient's diplopia. These features allow for the collection of baseline information on post-stroke gross and fine motor skills at a very early stage of recovery and provide the opportunity to potentially distinguish between the effects of ocular and cerebellar issues. Of note, both the results with healthy young adults and those with patients were found to be significant after only a fraction of the trials, indicating that the task could be substantially shortened and still provide a sufficiently precise measure of reaching accuracy. This rapid pace is particularly significant in the context of individuals with muscle weakness who may not be able to sustain activity for long periods of time.

Our results also show that even over a limited number of trials individuals with recent stroke demonstrate changes in their reaching accuracy, suggesting that this paradigm is sensitive to improvement or deterioration, critical for use in rehabilitative training. Of note, we detected a dissociation between the amount of fatigue in the isolated proprioception trials and the visual-proprioceptive integration trials in one of the stroke patients. The ability to measure these differences offers exciting opportunities to learn more about how specific sensory properties are affected by stroke. Moreover, the back-and-forth reaching design of our experiments mimics a clinical evaluation of motor coordination called the finger-to-nose test. By evaluating a patient's ability to quickly and accurately reach for both an externally-referenced target (the administrator's finger) and a self-referenced target (the patient's nose), this clinical test serves as rapid yet imprecise way to measure coordination. Many clinicians use the finger-to-nose test to measure upper-body coordination over the course of stroke recovery [75,76], but it remains a subjective tool with limited external validity. Using our VR paradigm, these same fine motor skills can be assessed in a way that provides detailed measurements without the need of a trained clinician to administer a coordination assessment.

As preliminary work, this study contains several limitations. While there are many benefits to the flexibility of a VR experience, the self-guided nature of it does introduce some differences in the stimulus presentation from person to person. This technique achieves more realistic interactions in a less repetitive and predictable environment, but does somewhat decrease the degree of control the experimenter has over the consistency of the experience. The experiments detailed above were self-paced, meaning that some participants could choose to move quickly and may be prone to greater errors while others could choose to take their time and demonstrate higher accuracy. Future work in which rate of action is a concern can employ a system to artificially pace the participant could be introduced. With this present study however, because each participant served as their own control and the trials of the two conditions in each experiment were randomly intermixed, we believe that the differences between conditions remains a valid metric of accuracy differentiation on an individual basis. This single-subject design also accounts for any variability in familiarity with VR, which otherwise could have

provided an advantage to those who have used VR in the past. The technology itself also has limitations, as the hand tracking accuracy has limitations associated with camera frame rate and figure/ground segmentation issues. These problems could cause gaps in tracking that may influence results, but the environment was well-lit and kept clear of objects that would interfere with tracking to reduce these confounds during each experiment.

Our sample size of adults with recent cerebellar stroke is small and is not representative of the wide variability of motor and visual complications that can be caused by a stroke. Our feasibility experiment intends only to show that VR is sensitive, adaptable, can be used by individuals with a variety of limitations, and can be conducted at the bedside. The patient group is also solely comprised of older adults, indicating that at this stage limited conclusions can be made about the role of recent stroke because age is a strong confounding factor. Future work should include a sample of healthy older adults who can be compared to the group of older adults with recent stroke to evaluate accuracy and learning differences.

## Conclusion

This paper highlights the promising application of commercially available virtual reality headsets to efficiently study perceptual and motor processing during naturalistic hand movements. Differences in reaching accuracy in various conditions were measurable in a short amount of time with very few trials. By studying the action-perceptual loop in a dynamic, multisensory environment, the field of psychophysics can move closer to understanding how perception varies across real-life settings. The adaptability and mobility of this equipment also offers opportunities to uncouple visual and proprioceptive cues to study the weighting and interaction of these domains in clinical populations in any setting. As affordable and accessible technology, future work incorporating additional participant groups and multisensory environments offers great potential to understand how different factors affect sensory processing.

## Supporting information

**S1 Video.**
(MP4)

**S2 Video.**
(MP4)

**S1 Data.**
(ZIP)

**S1 File. Video of example Visible/Invisible Hand experiment trials.**
(ZIP)

**S2 File. Video of example Memory Delay experiment trials.**
(ZIP)

## Acknowledgments

The authors would like to acknowledge Shui'Er Han, PhD., for her help resolving several VR coding issues and Paige Hepple for her help coordinating data collection with the stroke patients. The time and effort of ELI was supported by NIH T32 GM007356 during manuscript preparation.

## Author Contributions

**Conceptualization:** E. L. Isenstein, A. LoPrete, E. J. Knight, A. Busza, D. Tadin.

**Data curation:** E. L. Isenstein, T. Waz, Y. Hernandez, A. Busza, D. Tadin.

**Formal analysis:** E. L. Isenstein, D. Tadin.

**Investigation:** E. L. Isenstein, T. Waz, Y. Hernandez, A. Busza, D. Tadin.

**Methodology:** E. L. Isenstein, A. LoPrete, Y. Hernandez, E. J. Knight, A. Busza, D. Tadin.

**Project administration:** E. L. Isenstein, T. Waz, Y. Hernandez.

**Software:** E. L. Isenstein, A. LoPrete, E. J. Knight.

**Supervision:** A. Busza, D. Tadin.

**Validation:** D. Tadin.

**Visualization:** E. L. Isenstein, D. Tadin.

**Writing – original draft:** E. L. Isenstein, T. Waz, D. Tadin.

**Writing – review & editing:** E. L. Isenstein, T. Waz, A. LoPrete, Y. Hernandez, E. J. Knight, A. Busza, D. Tadin.

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
