## [Decision Letter · Decision Letter 0]

22 Jun 2022

PONE-D-22-07057Rapid assessment of hand reaching using virtual reality and application in cerebellar strokePLOS ONE

Dear Dr. Isenstein,

Thank you for submitting your manuscript to PLOS ONE. After careful consideration, we feel that it has merit but does not fully meet PLOS ONE’s publication criteria as it currently stands. Therefore, we invite you to submit a revised version of the manuscript that addresses the points raised during the review process.

The reviewers find this study to be interesting, however, one reviewer does not feel the issues raised in the introduction have been addressed in the study. Further, some clarification in the methods is needed, and concerns with the statistical analyses are noted. The authors should take into consideration the comments below.

We look forward to receiving your revised manuscript.

Kind regards,

Krista Kelly, Ph.D.

Academic Editor

PLOS ONE

Journal Requirements:

Additional Editor Comments (if provided):

1) Why was timing information not included? Impaired performance can be shown in more than just accuracy, for example reaction time (time to reach onset, time to reach dot etc..) as well as precision (noted by Reviewer 2).

2) The figures provided are pixelated and hard to read. Please upload ones that adhere to the PLOS ONE policies.

3) How was sample size calculated? Did the authors have enough power with 20 participants? Further, there is no statistical analysis section. Please add a section on power and statistical analyses to the methods.

4) The reviewers had issues with the statistical analyses that need to be addressed.

Reviewers' comments:

Reviewer's Responses to Questions

**Comments to the Author**

1. Is the manuscript technically sound, and do the data support the conclusions?

Reviewer #1: Yes

Reviewer #2: Partly

2. Has the statistical analysis been performed appropriately and rigorously? 

Reviewer #1: I Don't Know

Reviewer #2: No

3. Have the authors made all data underlying the findings in their manuscript fully available?

Reviewer #1: Yes

Reviewer #2: Yes

4. Is the manuscript presented in an intelligible fashion and written in standard English?

Reviewer #1: Yes

Reviewer #2: Yes

5. Review Comments to the Author

Reviewer #1: This is an interesting paper - the introduction raises some important issues but I don't necessarily feel these issues have been addressed in the study. I'm still not exactly sure how the research advances the field - I think this is most that it has not been clearly articulated and in a way that makes sense to lay readers. Some specifics:

1. The introduction is very long and is under referenced. Could it be a little more concise (and references add to support statements where needed)

2. The abstract lacks detail on the design of the studies and number of participants

3. Use of the term 'subjects' is not ideal - should be participants

4. The research question (at the end of the intro is not specific) - how is feasibility defined? this is important given specificity of the research methods

5. Results (number, characteristics of participants) are presented in the 'methods'

6. There are several figures presented which are hard to interpret - perhaps a more selective approach and better explanation would help. I am not able to assess the statistics.

Reviewer #2: Review of PONE-D-22-07057

Summary

The authors aim at testing the feasibility of a Virtual Reality system in assessing reaching performance, both in healthy adults and in pathological populations. The study is designed around a relatively standard paradigm about reaching accuracy, in which participants were to reach to a visual target, either with vision of their virtual hand or not, and either with online visibility of the target or not (memory-guided). In addition, a short version of the first task was applied to two stroke patients. The results show the well-established phenomenon that reach endpoint accuracy is poorer when the moving hand and/or the target is invisible. However, the authors want to emphasize as the main novelty of this work, the feasibility of using such VR systems for easy, portable, and reliable measures of human fine motor skills.

Main comments

The study is well-written, easy to follow, and makes its contributions explicit. It is technically sound, the performed analyses make sense, and the results are plausible. I do not have many comments, nor I feel that any of these require particular attention before recommending to the editor that this manuscript can be part of PLoS One table of contents. I would nevertheless encourage the authors to address the following issues.

1. The assessment of the virtual reality tool is based on the measurement of endpoint constant error (accuracy). This is a common measure in visuomotor research, and captures well the effects of visual information on manual control of action. I would have expected that the authors would assess the tools in a bit more detail though. Accompanying measures of endpoint accuracy is often that of endpoint precision (variable error), which may or may not follow the same pattern of endpoint accuracy (e.g., Monaco et al., 2010, Exp Brain Res). Considering the available data, I believe that reporting about endpoint precision (e.g., area and orientation of 95 CI) in the manuscript would meaningfully enrich the contents of the manuscript.

2. The authors base much of their motivation for this study on the premise that typical experiments impose relatively simple stimuli and restricted scenarios (e.g., lines 64-66), implying that these experiments are far from real-world activities, which is the gap that Virtual Reality can close. Though this is partly true, recently there is a growing number of studies implementing real-world naturalistic experiments. Examples include the recording of body and gaze when walking in the nature (e.g., Matthis et al., 2018, Curr Biol; Valsecchi et al., 2020, ACM in Eye Tracking), or when grasping objects (e.g., Land & Hayhoe, 2001, Vis Res; Voudouris et al., 2019, JoV), just to name a few examples. Without arguing against the importance of assessing Virtual Reality tools on measuring human behavior, the authors should acknowledge that research on the field was not only typically constrained, but that it has also been expanding in real-world settings.

3. Regarding the task itself, I have two main comments:

First, as seen in the supplementary videos, the recorded participant is remarkably accurate, considering that the target sphere ‘explodes’ (hit) in every single trial, even when the moving hand or the target sphere is invisible. Was the target sphere always exploding, no matter whether the participant correctly hit its center? Or is the demonstrated participant a well-trained one? This should be clarified.

Second, the endpoint error is measured as the distance between the center of the sphere and the position of the index finger at the moment when the finger crossed the 60-deg arc. Did the authors consider both remaining directions (lateral error and vertical error) when calculating this error or not?

4. Regarding the analysis, I am not convinced that measuring the p-value as the trials develop is a useful method. First of all, running constantly statistical tests can lead to false positives (even more so when the authors do not even correct for the abundant number of statistical tests). The authors should first explain what is the underlying reason for examining this aspect –is there an underlying hypothesis, which might be addressed in another way? From how I understand the development of the authors’ rationale, their aim is to see at what stage the experimental effect is systematic/robust, so that they can suggest how much the experiment can be reduced with respect to the number of trials and therefore to the time required. If this is the case, running too many statistical tests may also cause the impression that a few trials are necessary to reproduce the effect, but this impression may be false due to a type I error.

An alternative idea would be to split each block of trials performed by each subject in smaller subblocks of, say, 25 trials each, and then run a one-way ANOVA with four levels (epochs) to test whether these change over time. In any case, I recommend removing the analysis about the temporal evolution of the p-value from all sections of the manuscript.

5. More specific comments

Line 133: muscle contraction and visual perception in the same sentence, next to each other, read somewhat odd. It feels there should be a connection between the two, but I do not see any apparent one. Perhaps some revision would help here.

Line 138: How ‘small’ is ‘small’? Writing this sample size in a more explicit way will make the sentence more straightforward.

Line 195: “…they approached the periphery”. When periphery was approached with their hands or with some other part of their body?

Lines 220-221: At what distance was the target presented? From later information, it appears that this was at arm length, but I think it should be made more explicit here already.

Methods: Please mention what statistical tests are being used and whether there were normality checks supporting the use of parametric testing.

Results: Reporting the p-values in a more conventional way would probably facilitate readability.

Discussion: I think some parts can be shortened or merged. For instance, the contents of paragraph starting in line 470 made me think in what way VR can be used effectively to measure endpoint accuracy. Is it based on feasibility (e.g., portability, comfort, battery…) or on the fact that endpoint accuracy values reproduce previous experimental work? Reading the paragraph, it becomes apparent that the former is the correct answer, but this aspect is then back into discussion a couple of paragraphs later (e.g., starting at line 501). I feel that this paragraph is too detailed and does not contribute much, rather distracting from the main messages. One idea would be to merge such instances in more concise piece.

6. PLOS authors have the option to publish the peer review history of their article (what does this mean?). If published, this will include your full peer review and any attached files.

Reviewer #1: No

Reviewer #2: No

---

## [Author Response · Author response to Decision Letter 0]

27 Jul 2022

Thank you for you thoughtful and valuable comments on our manuscript. We have addressed the comments below:

Editor:

1) Why was timing information not included? Impaired performance can be shown in more than just accuracy, for example reaction time (time to reach onset, time to reach dot etc..) as well as precision (noted by Reviewer 2).

- We welcome the opportunity include additional data to support our findings. Data on the timing of the reaching has been added for Experiments 1 and 3, with no significant difference between the visible and invisible reaching times in either the healthy control group or the stroke group. These results show that the amount of time spent reaching is not driving the differences in accuracy between the two conditions. Unfortunately, this data is not available for Experiment 2 due to an issue with data collection. In addition, data reaching precision (as also noted by Reviewer 2) has been added for all three experiments. These findings show a significant effect of precision in Experiment 1 (p = .04) and in Experiment 2 (p < .01). Of note, we also found significant correlations between precision and accuracy in both Experiment 1 (visible p < .01; invisible p =.02) and Experiment 2 (standard p = .04; delayed p < .01).

2) The figures provided are pixelated and hard to read. Please upload ones that adhere to the PLOS ONE policies.

- Thank you for bringing this to our attention. We changed the file type and this seems to have resolved the issue.

3) How was sample size calculated? Did the authors have enough power with 20 participants? Further, there is no statistical analysis section. Please add a section on power and statistical analyses to the methods.

- At your suggestion, a specific section on statistical analysis has been added to make the statistical information clearer and more accessible – thank you for indicating that this would benefit our paper. Because this was a novel paradigm and prior data of this nature was not available, we were unable to initially conduct a power analysis. As such we selected 18-20 as our sample size because if greater than 20 participants were needed to find a significant result then the utility of our approach would be questionable.

- 

- We have now also conducted post-hoc power analyses to demonstrate the statistical power of our results; as we expected, power was high. In Experiment 1, post-hoc power analysis to achieve a power of .8 yielded a minimum sample of 3 when all 100 trials are included and a minimum sample of 8 when only the first 25 trials were included. In Experiment 2, post-hoc power analysis to achieve a power of .8 yielded a minimum sample of 4 when all 100 trials were included a minimum sample of 11 when only the first 25 trials were included. However, we opted to not include these post-hoc power analyses in the final manuscript because it does not offer new information about the credibility of the results.

- 

4) The reviewers had issues with the statistical analyses that need to be addressed.

- Concerns regarding the statistical analysis have been addressed by removing the cumulative p value plots. Instead, we illustrate the robustness of our approach by including a secondary analysis on just the first 25 trials in each condition. Figures 2 and 5 now include bar and line plots showing the group and individual level results after only 25 trials. In the text we also detail that in Experiment 1 after 25 trials the p value comparing visible and invisible hand reaching accuracy was < .001, in Experiment 2, the p value comparing standard and delayed reaching was < .01, and in Experiment 3 patient one had a p value of < .001 and patient 2 had a p value of .04. 

Reviewer 1:

This is an interesting paper - the introduction raises some important issues but I don't necessarily feel these issues have been addressed in the study. I'm still not exactly sure how the research advances the field - I think this is most that it has not been clearly articulated and in a way that makes sense to lay readers. Some specifics:

1. The introduction is very long and is under referenced. Could it be a little more concise (and references add to support statements where needed)

- Per the reviewer’s suggestion, we have made the introduction clearer and more concise, and agree that these changes improve the overall flow and clarity of the paper. Additional references have been added throughout to support the background and context of the introduction. 

2. The abstract lacks detail on the design of the studies and number of participants

- Thank you for catching this oversight - additional detail on the design and size of the experiments has been added to the abstract. 

3. Use of the term 'subjects' is not ideal - should be participants

- The term ‘subject’ has been replaced by ‘participant’ throughout the manuscript.

4. The research question (at the end of the intro is not specific) - how is feasibility defined? this is important given specificity of the research methods

- We appreciate the reviewer’s inquiry regarding the overarching goal of the paper, to gauge the feasibility of this kind of research. The research question, whether we can use virtual reality technology to assess fine motor skills in healthy adults as well as those with recent stroke, has been clarified at the end of the introduction.

5. Results (number, characteristics of participants) are presented in the 'methods'

- We have moved the key participant information to the beginning of the Results section.

6. There are several figures presented which are hard to interpret - perhaps a more selective approach and better explanation would help. I am not able to assess the statistics.

- Please see the fourth response to editor comments above for a full response in how we have made the figures more accessible. In short, the cumulative p value plots have been removed and instead we have added plots and analyses focusing on the accuracy and precision after only 25 trials. 

Reviewer 2:

Summary

The authors aim at testing the feasibility of a Virtual Reality system in assessing reaching performance, both in healthy adults and in pathological populations. The study is designed around a relatively standard paradigm about reaching accuracy, in which participants were to reach to a visual target, either with vision of their virtual hand or not, and either with online visibility of the target or not (memory-guided). In addition, a short version of the first task was applied to two stroke patients. The results show the well-established phenomenon that reach endpoint accuracy is poorer when the moving hand and/or the target is invisible. However, the authors want to emphasize as the main novelty of this work, the feasibility of using such VR systems for easy, portable, and reliable measures of human fine motor skills.

Main comments

The study is well-written, easy to follow, and makes its contributions explicit. It is technically sound, the performed analyses make sense, and the results are plausible. I do not have many comments, nor I feel that any of these require particular attention before recommending to the editor that this manuscript can be part of PLoS One table of contents. I would nevertheless encourage the authors to address the following issues.

1. The assessment of the virtual reality tool is based on the measurement of endpoint constant error (accuracy). This is a common measure in visuomotor research, and captures well the effects of visual information on manual control of action. I would have expected that the authors would assess the tools in a bit more detail though. Accompanying measures of endpoint accuracy is often that of endpoint precision (variable error), which may or may not follow the same pattern of endpoint accuracy (e.g., Monaco et al., 2010, Exp Brain Res). Considering the available data, I believe that reporting about endpoint precision (e.g., area and orientation of 95 CI) in the manuscript would meaningfully enrich the contents of the manuscript.

- Thank you for this insightful comment. We agree that adding endpoint precision as a metric of performance will substantially strengthen the paper, and we have added this information for all three experiments, with the results detailed in the first response to the editor. However, we unfortunately do not have information on the orientation of the reaching error, as only the absolute error regardless of direction was recorded. 

2. The authors base much of their motivation for this study on the premise that typical experiments impose relatively simple stimuli and restricted scenarios (e.g., lines 64-66), implying that these experiments are far from real-world activities, which is the gap that Virtual Reality can close. Though this is partly true, recently there is a growing number of studies implementing real-world naturalistic experiments. Examples include the recording of body and gaze when walking in the nature (e.g., Matthis et al., 2018, Curr Biol; Valsecchi et al., 2020, ACM in Eye Tracking), or when grasping objects (e.g., Land & Hayhoe, 2001, Vis Res; Voudouris et al., 2019, JoV), just to name a few examples. Without arguing against the importance of assessing Virtual Reality tools on measuring human behavior, the authors should acknowledge that research on the field was not only typically constrained, but that it has also been expanding in real-world settings.

- We appreciate this important consideration being pointed out. Several studies have now been referenced at the end of the first paragraph of the introduction exemplifying other work that has been done to conduct research in a more naturalistic setting.

3. Regarding the task itself, I have two main comments:

First, as seen in the supplementary videos, the recorded participant is remarkably accurate, considering that the target sphere ‘explodes’ (hit) in every single trial, even when the moving hand or the target sphere is invisible. Was the target sphere always exploding, no matter whether the participant correctly hit its center? Or is the demonstrated participant a well-trained one? This should be clarified.

Second, the endpoint error is measured as the distance between the center of the sphere and the position of the index finger at the moment when the finger crossed the 60-deg arc. Did the authors consider both remaining directions (lateral error and vertical error) when calculating this error or not?

- We have clarified that the target explodes regardless of where along the 60 degree arc the finger passes through, not just when the target itself was touched. We now also directly mention that the reaching error captures both vertical and horizontal error. 

4. Regarding the analysis, I am not convinced that measuring the p-value as the trials develop is a useful method. First of all, running constantly statistical tests can lead to false positives (even more so when the authors do not even correct for the abundant number of statistical tests). The authors should first explain what is the underlying reason for examining this aspect –is there an underlying hypothesis, which might be addressed in another way? From how I understand the development of the authors’ rationale, their aim is to see at what stage the experimental effect is systematic/robust, so that they can suggest how much the experiment can be reduced with respect to the number of trials and therefore to the time required. If this is the case, running too many statistical tests may also cause the impression that a few trials are necessary to reproduce the effect, but this impression may be false due to a type I error.

An alternative idea would be to split each block of trials performed by each subject in smaller subblocks of, say, 25 trials each, and then run a one-way ANOVA with four levels (epochs) to test whether these change over time. In any case, I recommend removing the analysis about the temporal evolution of the p-value from all sections of the manuscript.

- Please see the fourth response to editor comments above for a full response in how we have made the figures more accessible. In short, the cumulative p value plots have been removed and instead we have added plots and analyses focusing on the accuracy and precision after only 25 trials, showing that the total experiment could be a quarter of the original length and still be sensitive enough to the differences between conditions. 

5. More specific comments

Line 133: muscle contraction and visual perception in the same sentence, next to each other, read somewhat odd. It feels there should be a connection between the two, but I do not see any apparent one. Perhaps some revision would help here.

Line 138: How ‘small’ is ‘small’? Writing this sample size in a more explicit way will make the sentence more straightforward. - REMOVED

Line 195: “…they approached the periphery”. When periphery was approached with their hands or with some other part of their body?

Lines 220-221: At what distance was the target presented? From later information, it appears that this was at arm length, but I think it should be made more explicit here already.

- Thank you for such helpful and specific comments on individual pieces of language throughout the paper! The have all been addressed and revised according to your suggestions.

Methods: Please mention what statistical tests are being used and whether there were normality checks supporting the use of parametric testing.

- We have also added a specific section for statistical testing to make it clear what is being done. All data has been checked for normality and due to the presence of least one non-normal condition in each experiment, Wilcoxon Signed Rank Tests have been used to replace the t tests. These changes have not affected the patterns of significance previously reported.

Results: Reporting the p-values in a more conventional way would probably facilitate readability.

- Please see the above responses regarding how the p values are now being reported. 

Discussion: I think some parts can be shortened or merged. For instance, the contents of paragraph starting in line 470 made me think in what way VR can be used effectively to measure endpoint accuracy. Is it based on feasibility (e.g., portability, comfort, battery…) or on the fact that endpoint accuracy values reproduce previous experimental work? Reading the paragraph, it becomes apparent that the former is the correct answer, but this aspect is then back into discussion a couple of paragraphs later (e.g., starting at line 501). I feel that this paragraph is too detailed and does not contribute much, rather distracting from the main messages. One idea would be to merge such instances in more concise piece.

- We agree that the discussion was somewhat circuitous, and it has now been slightly restructured to make it more concise and clear about what we want the main takeaways from the paper to be. Thank you for your advice.

---

## [Decision Letter · Decision Letter 1]

18 Aug 2022

PONE-D-22-07057R1Rapid assessment of hand reaching using virtual reality and application in cerebellar strokePLOS ONE

Dear Dr. Isenstein,

Thank you for submitting your manuscript to PLOS ONE. After careful consideration, we feel that it has merit but does not fully meet PLOS ONE’s publication criteria as it currently stands. Therefore, we invite you to submit a revised version of the manuscript that addresses the points raised during the review process. Specifically, the reviewer is requesting more of a discussion on action in the Introduction, and clarification for some of the Methods and Results regarding the direction of the error that is recorded and the reaching time. Other minor suggestions are provided.

We look forward to receiving your revised manuscript.

Kind regards,

Krista Kelly, Ph.D.

Academic Editor

PLOS ONE

Journal Requirements:

Reviewers' comments:

Reviewer's Responses to Questions

**Comments to the Author**

1. If the authors have adequately addressed your comments raised in a previous round of review and you feel that this manuscript is now acceptable for publication, you may indicate that here to bypass the “Comments to the Author” section, enter your conflict of interest statement in the “Confidential to Editor” section, and submit your "Accept" recommendation.

Reviewer #1: All comments have been addressed

Reviewer #2: (No Response)

2. Is the manuscript technically sound, and do the data support the conclusions?

Reviewer #1: Yes

Reviewer #2: Yes

3. Has the statistical analysis been performed appropriately and rigorously? 

Reviewer #1: I Don't Know

Reviewer #2: Yes

4. Have the authors made all data underlying the findings in their manuscript fully available?

Reviewer #1: Yes

Reviewer #2: Yes

5. Is the manuscript presented in an intelligible fashion and written in standard English?

Reviewer #1: Yes

Reviewer #2: Yes

6. Review Comments to the Author

Reviewer #1: I have no additional feedback for the authors. They have addressed my previous feedback satisfactorily.

Reviewer #2: The authors have addressed all my comments and I believe that the manuscript now reads more focused and clearer. I still have some remaining comments, some of which arise due to the revisions whereas a few others concern issues that I could have spotted in the first round. Apologies for not bringing these up earlier. All lines mentioned below concern the track-changes document.

Abstract: the first lines of the abstract greatly focus on the ‘perception’ side and the limitations of the traditional experiments on perception. Though this true, this study here focuses more on action/behavior. I would encourage the authors to either bring the ‘action’ part more explicitly, or even discuss the perception-action part altogether (than only talking about ‘perception’).

Introduction, paragraph 1 (lines 55-68): Similarly here, the focus appears to be on ‘perception’. Yes, perception research started with rather passive, highly controlled designs, but this has now evolved to more naturalistic experiments. In the same line, ‘action’ research started with constrained paradigms, such as the studies of Jeannerod on grasping, but has evolved to whole-body tracking in the nature, as the authors also cite the work of Matthis and colleagues. Therefore, as in the abstract, I recommend also here to focus also on action, not only on perception.

Methods, lines 149-154: Please introduce the participants of experiment 3 somewhere here. This is important otherwise the term ‘healthy’ in line 152 reads odd, while doing so will facilitate reading of the subsequent section (e.g., lines 161-162).

Line 207: It remains unclear whether the error is measured only on the lateral direction or the vertical direction is also accounted. Please make this more explicit.

Line 209: did the reaching time indeed consider the moment when the participant ‘touched the target’? In other instances (e.g., lines 199-201), it is mentioned that the trial ended when the participant crossed the arc, implying that this would be used as the reaching time. If reaching time is indeed measured based on when the participant touched the target, what happened in trials in which the participant did not touch the target? If the reaching time is measured based on the time when the participant crossed the arc, this line needs to be revised (as also other instances throughout the manuscript, for instance line 232, 262, and possibly elsewhere).

Lines 226-227: Could you please clarify whether the 100 trials of one of the conditions were presented before/after the 100 trials of the other condition, or whether the two conditions were randomly presented randomly interleaved across the 200 trials?

Line 276: “…of their accuracy”. I think the authors here should state “…of their endpoints”. Also, is precision calculated as the SD along the lateral direction only? Please clarify in the manuscript.

Lines 309-312: These here read as ‘methods’ but are presented in the ‘results’. My recommendation is to define the three measures (accuracy, precision, reaching time) in the Methods, one after the other, so that it easier for the reader to follow the analysis.

Line 326: “of this measure”. Which measure? The previous part refers to two different variables, accuracy and precision. Why not calculate precision and reaching time also on the basis of the first 25 trials? Then the authors, for each experiment, would have a first paragraph with results about the three measures (accuracy, precision, time) considering all trials, and a second paragraph with the respective results when considering only the first 25 trials. Then the results would be easier to follow and interpret.

If the authors would like to focus mainly on accuracy, this should become more explicit, ideally with a reason why.

Lines 333-335: Does this refer to reaching time considering all trials or only the first 25 trials of each condition?

7. PLOS authors have the option to publish the peer review history of their article (what does this mean?). If published, this will include your full peer review and any attached files.

Reviewer #1: No

Reviewer #2: No

---

## [Author Response · Author response to Decision Letter 1]

8 Sep 2022

Thank you for the additional attentive comments on our manuscript. We have addressed the comments below:

Reviewer #2: 

1. The authors have addressed all my comments and I believe that the manuscript now reads more focused and clearer. I still have some remaining comments, some of which arise due to the revisions whereas a few others concern issues that I could have spotted in the first round. Apologies for not bringing these up earlier. All lines mentioned below concern the track-changes document.

a. Thank you again for the valuable comments, we agree that the paper is more streamlined and easier to follow. 

2. Abstract: the first lines of the abstract greatly focus on the ‘perception’ side and the limitations of the traditional experiments on perception. Though this true, this study here focuses more on action/behavior. I would encourage the authors to either bring the ‘action’ part more explicitly, or even discuss the perception-action part altogether (than only talking about ‘perception’).

a. We have expanded the beginning of the abstract to emphasize the relevance of both perception and action to this manuscript.

3. Introduction, paragraph 1 (lines 55-68): Similarly here, the focus appears to be on ‘perception’. Yes, perception research started with rather passive, highly controlled designs, but this has now evolved to more naturalistic experiments. In the same line, ‘action’ research started with constrained paradigms, such as the studies of Jeannerod on grasping, but has evolved to whole-body tracking in the nature, as the authors also cite the work of Matthis and colleagues. Therefore, as in the abstract, I recommend also here to focus also on action, not only on perception.

a. The introduction now contains additional mention of the interplay of action and perception as well as past limitations of action research, such as small or restricted movements.

4. Methods, lines 149-154: Please introduce the participants of experiment 3 somewhere here. This is important otherwise the term ‘healthy’ in line 152 reads odd, while doing so will facilitate reading of the subsequent section (e.g., lines 161-162).

a. An introduction to the participants in Experiment 3 has been added at the beginning of the Methods section.

5. Line 207: It remains unclear whether the error is measured only on the lateral direction or the vertical direction is also accounted. Please make this more explicit.

a. Further clarification has been added that the error accounts for error in both the horizontal and vertical direction.

6. Line 209: did the reaching time indeed consider the moment when the participant ‘touched the target’? In other instances (e.g., lines 199-201), it is mentioned that the trial ended when the participant crossed the arc, implying that this would be used as the reaching time. If reaching time is indeed measured based on when the participant touched the target, what happened in trials in which the participant did not touch the target? If the reaching time is measured based on the time when the participant crossed the arc, this line needs to be revised (as also other instances throughout the manuscript, for instance line 232, 262, and possibly elsewhere).

a. Thank you for pointing out this discrepancy, we have clarified throughout the paper that the reaching time begins when the target appears and end when the index finger passes through the arc, rather than when it specifically touches the target.

7. Lines 226-227: Could you please clarify whether the 100 trials of one of the conditions were presented before/after the 100 trials of the other condition, or whether the two conditions were randomly presented randomly interleaved across the 200 trials?

a. We have clarified that the two conditions of trials are randomly interspersed together in all experiments. 

8. Line 276: “…of their accuracy”. I think the authors here should state “…of their endpoints”. Also, is precision calculated as the SD along the lateral direction only? Please clarify in the manuscript.

a. We have clarified here that endpoint accuracy is the metric being discussed. 

9. Lines 309-312: These here read as ‘methods’ but are presented in the ‘results’. My recommendation is to define the three measures (accuracy, precision, reaching time) in the Methods, one after the other, so that it easier for the reader to follow the analysis.

a. We have reorganized this information so the beginning of the Statistical Analysis section now defines the three metrics of reaching accuracy, reaching precision, and reaching time together. 

10. Line 326: “of this measure”. Which measure? The previous part refers to two different variables, accuracy and precision. Why not calculate precision and reaching time also on the basis of the first 25 trials? Then the authors, for each experiment, would have a first paragraph with results about the three measures (accuracy, precision, time) considering all trials, and a second paragraph with the respective results when considering only the first 25 trials. Then the results would be easier to follow and interpret.

If the authors would like to focus mainly on accuracy, this should become more explicit, ideally with a reason why.

a. Thank you for pointing out this unclear phrasing; we have clarified that we referred to the sensitivity of the experiment to assess reaching accuracy. We have also stated that reaching accuracy was the primary outcome measure because it has the greatest clinical significance and effect on quality of life and independence.

11. Lines 333-335: Does this refer to reaching time considering all trials or only the first 25 trials of each condition?

a. This line has been moved to clearly indicate that it is referring to reaching time across all 100 trials.

---

## [Editor Report · Decision Letter 2]

13 Sep 2022

Rapid assessment of hand reaching using virtual reality and application in cerebellar stroke

PONE-D-22-07057R2

Dear Dr. Isenstein,

We’re pleased to inform you that your manuscript has been judged scientifically suitable for publication and will be formally accepted for publication once it meets all outstanding technical requirements.

Kind regards,

Krista Kelly, Ph.D.

Academic Editor

PLOS ONE

Additional Editor Comments (optional):

No further comments.
---

## [Editor Report · Acceptance letter]

20 Sep 2022

PONE-D-22-07057R2 

Rapid assessment of hand reaching using virtual reality and application in cerebellar stroke 

Dear Dr. Isenstein:

I'm pleased to inform you that your manuscript has been deemed suitable for publication in PLOS ONE. Congratulations! Your manuscript is now with our production department. 

Kind regards, 

on behalf of

Dr. Krista Kelly 

Academic Editor

PLOS ONE